# Strip-Till One-Pass Technology in Central and Eastern Europe: A MZURI Pro-Til Hybrid Machine Case Study

**Iwona Jaskulska *** and **Dariusz Jaskulski**

Department of Agronomy, Faculty of Agriculture and Biotechnology, University of Science and Technology, 7 prof. S. Kaliskiego St., 85-796 Bydgoszcz, Poland; darekjas@utp.edu.pl
* Correspondence: jaskulska@utp.edu.pl

**Abstract:** The non-inversion tillage systems, including strip-till (ST), are the key element of conservation agriculture (CA). The aim of the 2012–2018 study has been to demonstrate the application of strip-till one-pass technology (ST-OP) on the farms of Central and Eastern Europe based on the use of Mzuri Pro-Til machines. There has also been an evaluation of the effect of that technology on the soil properties and the effects of crops growing. The scientific observations and field experiments were made, e.g., in Poland, Ukraine, Lithuania, the Czech Republic, Slovakia, the Eastern states of Germany, Belarus, Serbia, and Romania. ST-OP case study with the use of Mzuri Pro-Til machine can be applied for growing all the basic crops. Tillage with a simultaneous basic fertilization application and seeding made regularly for a few years in given field leads to favorable changes in the soil properties. As compared with the soil under conventional plough tillage (CT), the soil moisture, especially in the periods of rainfall deficit, the content of organic carbon and its fraction, the count of microorganisms and earthworms, as well as the enzymatic activity, are higher. This technology saves over 20–30 L ha$^{-1}$ of fuel, respectively, compared to reduced tillage (RT) and CT. Plant emergence is uniform, dense canopies and crop yields—not lower and even higher than for tillage and seeding commonly applied in Central and Eastern Europe. ST-OP can be thus an important element of field plant production as part of CA and sustainable development.

**Keywords:** agricultural machine; conventional tillage; crop yield; soil properties; strip tillage

## 1. Introduction

Tillage is the key factor affecting the soil in agroecosystems [1,2]. Depending on the tool design, especially the working parts of the tools, soil is inverted, loosened, compacted, mixed, and crushed [3,4]. The entire field surface or only its part, narrow strips, will be tilled. The tillage depth also differs. In some cases, tillage covers the entire soil thickness and it is, on average, 10–25 cm, and minimum 2–3 cm, namely at the seeding depth [5,6].

The sources of historic and contemporary literature demonstrate that tillage is subject to evolutionary changes along with the civilization development and scientific, technical, and technological progress [7,8]. The contemporary tillage methods and systems are adjusted to the habitat, social and economic conditions, and the agricultural policy in various regions of the world [9–11].

The CT dominates in the world and especially in Europe. From year to year, however, the acreage under RT and no-tillage (NT) increases, mostly in the regions threatened with erosion. Although the most up-to-date data are not available [12], in the second decade of the 21st century CT in Europe, except for a few countries, Bulgaria and Cyprus, was applied on more than 50% of the cropland. CA with RT is mostly used in South and North America. In those regions it covers more than 60% and

25% of the cropland, respectively. In Europe, including Poland, it is below 10% [13–15]. To protect against runoff, the water drop energy, the wind, the crop residue, the catch-crops biomass and mulch on the soil surface are used, preferably for NT and direct seeding (DS), as the key element of CA [16]. A low number of RT treatments is also essential [17,18]. Soil preparation and seeding are limited to a single treatment; soil loosening only with the sowing coulters. On the field surface there remains almost 100% of post-harvest residue [19,20]. The seedbed prepared in such way does not always ensure the conditions which would be favorable for seed germination and plant emergence, and the environmental benefits are identified only in some conditions [21,22]. A lack of deep tillage can limit the development of crop root systems, especially in heavy soils [23], whereas CT covers all the post-harvest residue. It is facilitated by seeding; a favorable space for root development is formed, however, soil is exposed to erosion [24], a decrease in the content of organic matter [25,26], and water losses [27]. Plough requires high inputs of energy, it generates high costs, and it has a negative effect on the economic result of the field plant production [28]. High differences in the impact of the CT system and NT on the soil, crops, and the environment inspire to search for new solutions. Over the last decades, the agricultural science and practice have been focusing on the ST method a lot [29].

ST combines the qualities of CT and DS. Plough tillage inverses and loosens the deep soil, it prepares favorable air–water–temperature conditions for growth, development, and physiological activity of the crop root systems. NT drastically limits the inputs and costs, it shortens the working time, and protects the environment. In agricultural practice, the top ten reasons for which that tillage system is gaining more and more importance are: considered to be a good seedbed preparation in narrow soil strips between which the soil is not loosened, fast soil warming-up in loosened strips, facilitating an early spring seeding, deep soil strip loosening to enhance the exchange of water and air, soil protection against erosion, precision fertilization, fertilization effectiveness, limiting the number of passes across the field, fuel consumption saving, cost saving, and a high yielding potential. These qualities are confirmed by the results of much of scientific research [30–32].

The successive agrotechnical practices, especially seeding, can be performed sometime after loosening the soil strips (ST two-pass) or during one machine pass (ST one-pass). The ST two-pass is used in wet soils. Soil strip loosening and aeration make the soil drying and warming easier. Seeding is performed after some time when the soil water and temperature conditions are favorable to seed germination. However, in general, ST-OP brings more economic, organizational, and environmental benefits [33]. ST, as compared with the CT, allows for limiting the inputs while maintaining the yield size, which was demonstrated for growing rice, by Islam et al. [34]. According to Cociu [35], the RT, including ST, facilitates a decrease in the fuel and labor inputs in corn, soya, and wheat production by more than 50% and reduces the costs while maintaining similar yields. The results of the studies reported by Šarauskis et al. [36] indicate that increasing the working depth from 0 to 200 mm increases the fuel consumption by 10.3–24.3% depending on the working speed. The lowest fuel consumption was recorded for the working speed of 2.5 m s$^{-1}$. The ST is more and more commonly applied both for growing plants in wide-spaced rows [37–39] and in narrow-spaced rows [40,41].

The ST machines have a few major working elements that affect the soil properties of the strips tilled, especially row cleaners, ripper teeth (coulters), discs, packer rollers. The working elements can be passive or active. The crop residue management, the width and depth of the soil strip loosened, the intensity of soil loosening and mixing, as well as fuel consumption, depend on the design, geometry, and the setting of the working elements and the working speed of the machine [42–44]. A technical problem is the construction of machines and their elements, especially for growing crops seeded at high density and in narrow-spaced rows, e.g., cereals. A loosened soil strip must be narrow and the width of non-loosened interrow is greater than in traditional seed drilling. For economic reasons, such machine should allow plant seeding for different, also wider, row spacing [45,46]. In the agricultural market, especially in Europe, such machines are few. An example of the universal hybrid machine operating compliant with the strip-till one-pass technology is Mzuri Pro-Til (Śmielin, Polska) [47,48].

This paper presents the functionality, design, basic technical parameters of Mzuri Pro-Til machines and the ST-OP followed with those machines. The aim of the field experiments and of the many-year scientific observations has also been to evaluate the effect of Mzuri Pro-Til machines and the ST-OP on the soil, crops, and the plant production results in the countries of Central and Eastern Europe. This technology was compared with CT and/or RT.

## 2. Materials and Methods

### 2.1. Hybrid Machine

Mzuri is a series of soil tillage machines. The most important position is the hybrid Pro-Till machines for belt soil loosening, fertilization, and sowing. Since 2018, the Mzuri Pro-Til machines (Figure 1) of British company Mzuri LTD have been manufactured in Poland; Mzuri-Agro sp. z o.o. sp. k. (Śmielin, Poland). The company is also a machinery distributor. The machine series includes the models with a varied working width, suspended and hooked, without and with a simultaneous application of mineral fertilizers, with a fixed and varied number of strips of the soil tilled (Table 1). The design of the sowing coulter allows for row seed drilling and precision drilling. As for row seed drilling, the seeds can be seeded in one or two rows per tilled soil strip. It is also possible to place the seeds not in a row but in a few-centimeter belt.

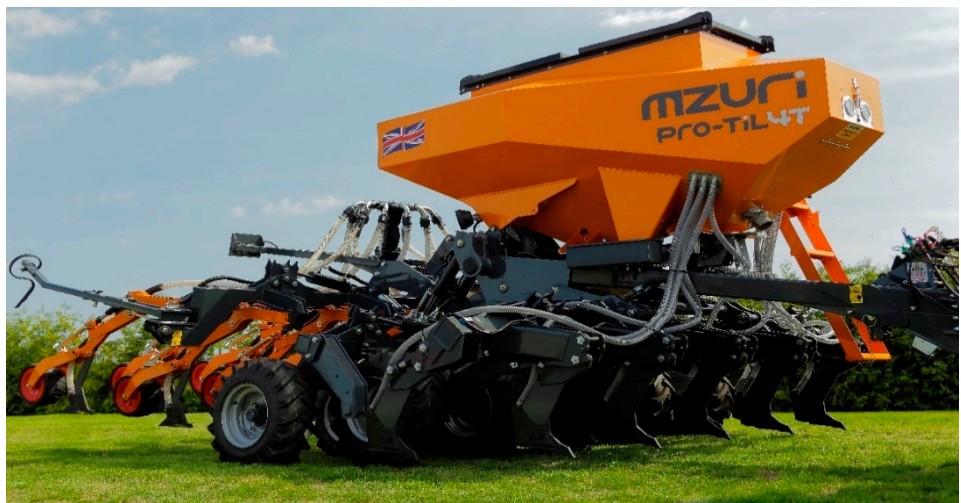

**Figure 1.** Mzuri Pro-Til machine used for tillage, fertilization, and seeding.

**Table 1.** Basic characteristics of the selected Mzuri Pro-Til models based on [47,49].

| Specification | PRO-TIL 3 | PRO-TIL 4 TSELECT | PRO-TIL 6 TSELECT | PRO-TIL 4 TXzact |
|---|---|---|---|---|
| Working width (m) | 3 | 4 | 6 | 4 |
| Hopper capacity (dm$^3$) | 1500 | 2800 | 4300 | 3400 |
| Dual Hopper (fertilizer/seeds) | no/yes | yes/yes | yes/yes | yes/yes |
| Row spacing (cm) | 33.3 | 36.4 or 72.8 | 35.0 or 70.0 | 36.4 or 72.8 |
| Number of rows | 9 | 11 or 6 | 17 or 9 | 11 or 6 |
| Precision sowing units | no | no | no | 6 |
| Tractor requirement (hp) | 150 | 220 | 300 | 200 |
| Working speed (km h$^{-1}$) | 6–15 | 6–15 | 6–15 | 6–15 |
| Aggregating with a tractor | linkage | trailed | trailed | trailed |

## 2.2. Study Area

Currently the biggest Mzuri Pro-Til machinery market is found in the countries of Central and Eastern Europe (Figure 2). Mzuri Pro-Til machines also operate in Great Britain, as well as in Western Europe, Scandinavia, Asia, New Zealand, and in Africa. Depending on the local environmental conditions and the specificity of agriculture, they are used to grow different crops (Figure 3). They allow for sowing the seeds varied in size, with different density, shallow and deep, at wide and narrow row spacing, in rows and in belts, irregular in a row and in precision drilling.

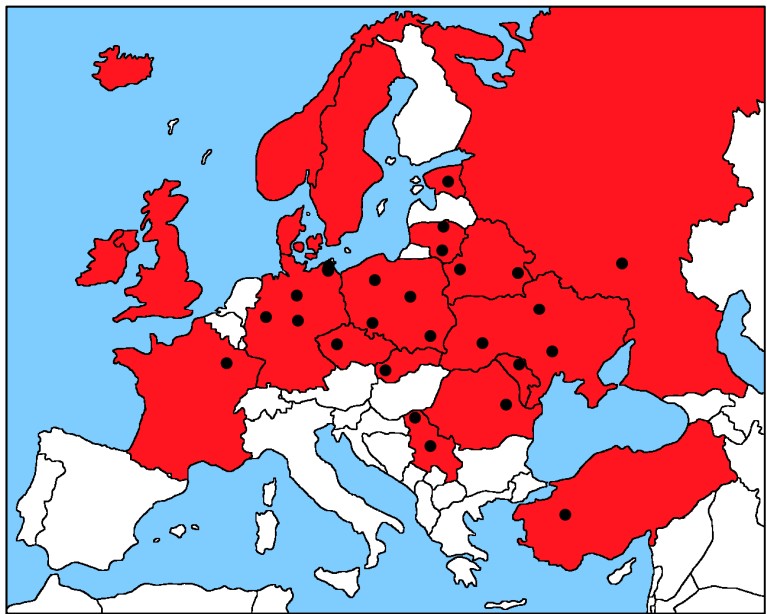

**Figure 2.** Area of operation (red) and Mzuri Pro-Til machines dealerships (●) in Europe. Europe outline map–source: https://paintingvalley.com/europe-map-sketch#europe-map-sketch-4.gif [50].

According to the Köppen climate classification, the experimental site is characterized by Dfb climate (cold, without dry season, warm summer) [51]. The soil characteristics in the fields of the experimental farm in Śmielin (Poland) are presented in the earlier works of the authors [41,52,53].

## 2.3. Strip-Till One-Pass Technology with Mzuri Pro-Til Machines

Mzuri Pro-Til machines facilitate the performance of many agrotechnical practices in one-pass (Figure 4). The front discs work first. They are found in front of the tines loosening the soil strips. The tines are set at a slight angle to the machine's movement direction. They cut long post-harvest residue or catch crop stems. Cutting the plant material allows for pushing it away into inter-rows and limits the risk of their accumulation in the machine tines. The tines with exchangeable wings in their lower part facilitate deep loosening of a narrow soil strip without bringing it up on the field surface, which secures the wet soil from fast drying, getting lumpy, and crusting. The back part of the tines is combined with the mineral fertilizer applicator. Profiling allows for placing the fertilizers in the entire loosened soil space, and partially even on its surface. Therefore, there is no risk of an excessive nutrient concentration in the zone of seed germination and sprout dying. Fertilizer is available to the roots, also to enhance their elongation during growth. Each loosening tine is followed by a pneumatic wheel for traction and, mostly, for soil pressing and compacting. It compacts the soil, removes the air pockets once it goes through the loosening tine. A hydraulic pressure adjustment for each wheel separately and a possibility to change the wheels' air pressure make the soil conditions in each row the same, irrespective of variation in the soil properties in the field.

Sowing coulters are entered into such soil strips. Each of them, separately suspended and equipped with a wheel, adjusts to the field unevenness, and ensures seed placement at the same depth. The exchangeable coulters and their working elements allow for single- and twin-row or belt drilling. The seeds are densely placed in the row, without spacing control, or precision seed drilled. The side wings, provided in some types of coulters, push away crop residues with soil from the seed row and profile the small ridges in inter-rows. Rows of plants occur in shallow furrows (Figure 5). Such field configuration after a pass of Mzuri Pro-Til enhances the absorption of higher amounts of water in the root growth zone, retaining the snow around the root collars and tillering nodes of the plants to help overwintering, as well as the accumulation of mineral fertilizer granules applied as top dressing in furrows. After a pass, the coulters without the side wings leave an even field, which allows for performing the interrow treatments and makes the crop harvest easier.

A rear press wheel is mounted behind the sowing coulter. It keeps the set seeding depth and acts as a pressing roller. It presses the soil around the seeds, which increases the water penetration. Mzuri Pro-Til machines are equipped with exchangeable rear press wheels that differ in design. Their selection depends on, e.g., the seeding type: single row, double row, or belt; the seeding depth; soil conditions: texture, moisture, a tendency to crusting. Behind the press wheels, as an option, there can be found a soil surface harrow-levelling element. Mzuri Pro-Til machines can be also equipped with the applicators of micro-granulated products (fertilizers, plant protection products) or catch-crop seeds.

The research of the results that have been presented here was performed using the Mzuri Pro-Til 4T machine with the working width of 4 m, equipped with 11 sections for loosening the soil strips and seed sowing. The distance between the loosening tines and the sowing coulters was 36.4 cm. In front of the loosening tines, there were mounted single flat disks, cutting and pushing the crop residue away. The width and the depth of the strip of the soil loosened was 12 cm and 20 cm, respectively. The sowing coulters were equipped with side wings. For cereals sowing, two-row coulters were used and for rapeseed sowing–one-row coulter. The working speed of the machine was 10 km h$^{-1}$.

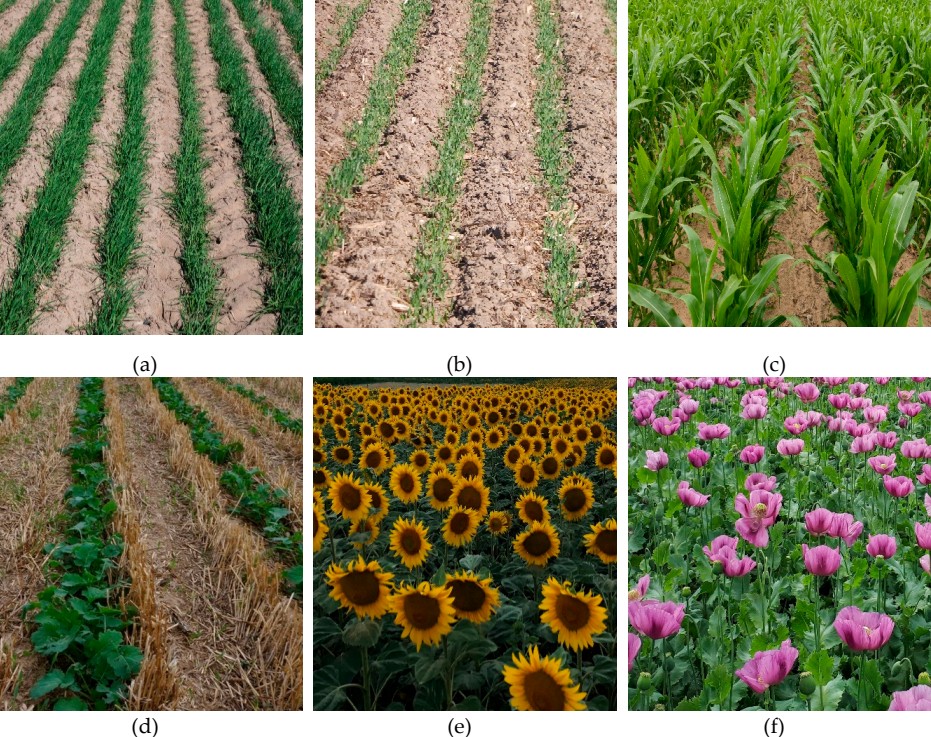

(a) (b) (c)

(d) (e) (f)

**Figure 3.** *Cont.*

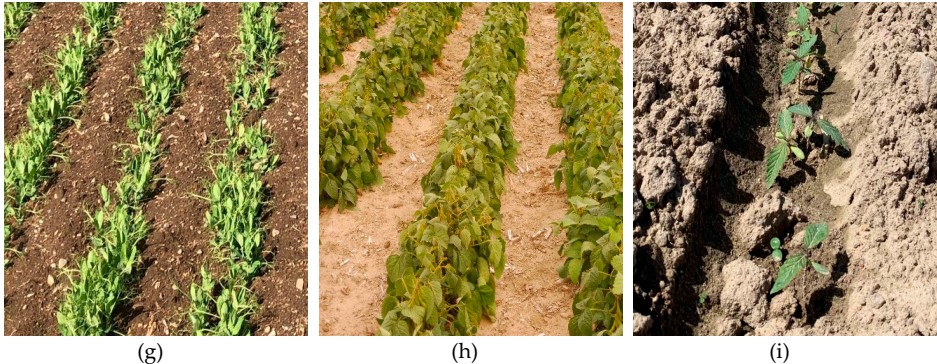

(g)                                       (h)                                       (i)

**Figure 3.** Crops sown in Central and Eastern Europe using Mzuri Pro-Til machines: (**a**) winter wheat (Czech Republic), (**b**) spring barley (Belarus), (**c**) maize (Serbia), (**d**) winter rapeseed (Lithuania), (**e**) sunflower (Romania), (**f**) opium poppy (Slovakia), (**g**) pea (Germany), (**h**) soybean (Ukraine), (**i**) hemp (Poland).

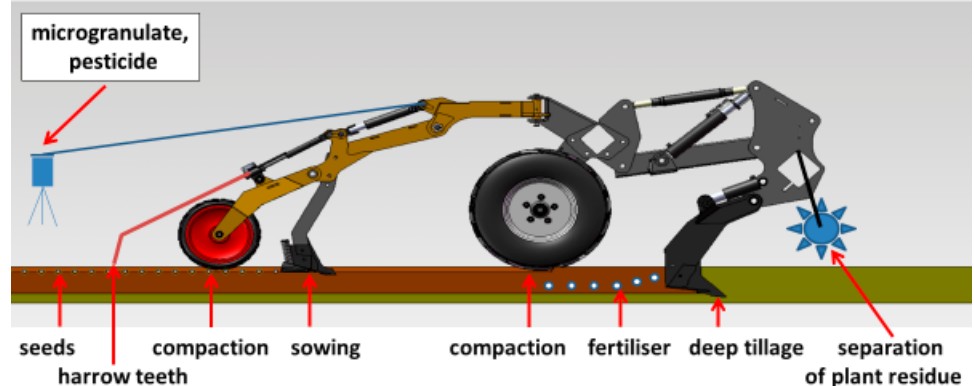

**Figure 4.** Diagram of the agrotechnical practices performed with the use of Mzuri Pro-Til machine.

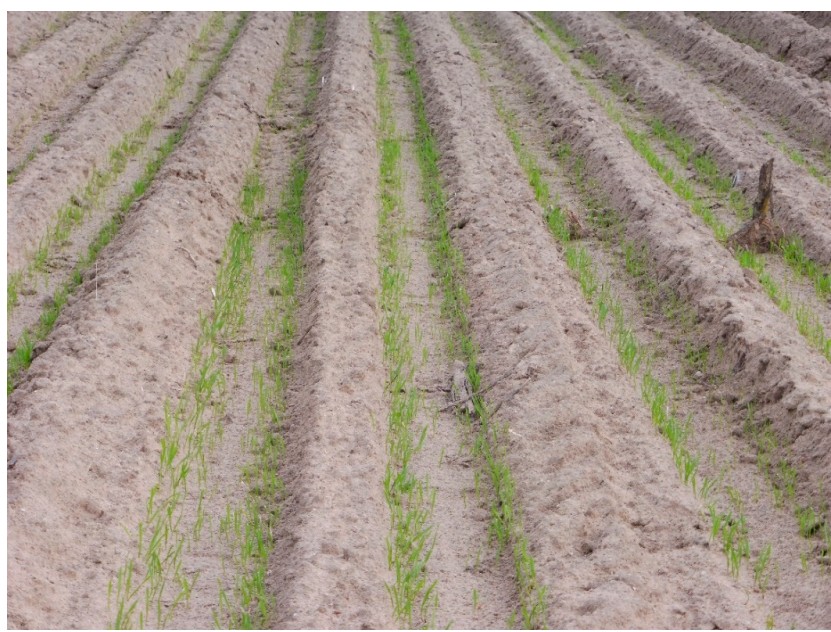

**Figure 5.** Field configuration after a Mzuri Pro-Til machine pass.

*2.4. Scientific Observations and Field Experiments*

The ST-OP was compared with CT and with CT and RT in two static experiments at Śmielin (Poland). The research and scientific observations in other countries of Central and Eastern Europe compared ST-OP and CT. The research focused on winter wheat, winter barley, and winter rapeseed. The plants were not sown on the same calendar date, but only on the day and density optimal for a given region. However, for the tillage and sowing treatments compared in the experiment (location), the other agrotechnical treatments were always the same. In many-year (2013–2019) static field experiments (Śmielin, Poland), and based on scientific observations and biometrics measurements on crop plantations during this period, emergence and plant wheat density (Poland, Ukraine, Lithuania, Czech Republic, Slovakia, Belarus), the following were determined and the methods described below:

1.  Soil moisture in spring–summer months of plant vegetation (volumetric soil water content directly in the field experiment—four points in each treatment).
2.  Soil moisture, bulk density, soil aggregate water stability (soil samples in laboratories), and penetration resistance in the field after 6 years of the application of Mzuri Pro-Til technology and CT. Soil sampling and measurements were performed at 10 points in the treatment.
3.  The total organic carbon and organic carbon fraction (in soil samples—ten points in each treatment after 6 years of different tillage methods).
4.  The count of microorganisms at 10 points in the treatment (the total count of bacteria, cellulolytic microorganisms, Actinobacteria, filamentous fungi), the activity of soil enzymes (dehydrogenase), the number of earthworms in the top layer of soil—after 6 years of field experiment.
5.  The evenness of winter wheat emergence in large-acreage fields (based on plant density on plantations over 10 ha). Measurement of plant density was performed at 15 points in the plantation.
6.  Fuel inputs on the performance of agrotechnical practices, especially tillage, pre-seeding application of fertilizers and the seeding of crops, on average, in the 6-year research period (register of fuel consumption by the tractor's on-board computer).
7.  Yield of winter crops (main crops in the area of study): rapeseed, wheat, barley yield determined from the entire surface of the experimental plots in four replications.

All the soil properties were determined in the 0–20 cm layer. The methodology for measuring and analyzing the experiments outlined above was as follows: (1) The volumetric soil water content over the plant vegetation period was monitored twice a month, using PR2/4 Profile Probe (Delta-T Devices Ltd, Burwell, UK), with the time domain reflectometry method in three successive years (2014–2016). The other physical soil properties were assayed 10 days after the winter rapeseed harvest in 2019. (2) The bulk density and moisture content (m/m) were measured following the soil sampling separately from the zone of rows and interrows in an undisturbed state to cylinders 100 cm$^3$ in size (Eijkelkamp Soil & Water, Giesbeek, NL). The soil penetration resistance was measured with hand penetrometer and soil aggregate water stability using Wet Sieving Apparatus (Eijkelkamp Soil & Water, Giesbeek, NL). (3) The content of organic carbon was assayed with Vario Max CN analyzer (Elementar, D, Hanau, Germany). The extractable organic carbon was assayed following the extraction of the soil sample with 0.004 M CaCl$_2$, at the soil to extractant ratio of 1:10. The fraction composition of organic carbon was assayed with the Schnitzer method. (4) The microorganisms in soil were counted on the inoculated and incubated media: YPS—total count of bacteria, Pochona—Actinobacteria, Martina with 30 μg mL$^{-1}$ streptomycin added—filamentous fungi, Congo—Red Agar CMC-Na—cellulolytic microorganisms. The solutions for microbiological cultures were made by adding 90 mL of Ringer's solution to 10 g of the soil sample. After the centrifugation for 30 minutes, the series of ten-time dilutions ($10^{-1}$ do $10^{-7}$) were made. The activity of dehydrogenase (DEH) in soil was assayed with the Thalmann [54] method, after the incubation of soil with 2,3,5-Triphenyltetrazolium chloride (TTC) and the colorimetric measurement of the absorbance of triphenylformazan (TPF) at 546 nm. The activity was expressed in mg TPF kg$^{-1}$ 24 h$^{-1}$. To evaluate the number of earthworms, soil monoliths 20 cm × 20 cm × 20 cm were sampled. The result was expressed in no m$^{-2}$ of the soil area. (5) The evenness of winter wheat

emergence was evaluated on big winter wheat production plantations in six countries. Plants were counted at 20 sites, which differed in terms of physiographic and soil properties. The field emergence was determined based on the following formula:

$$\text{Field emergence (\%)} = \text{PD/SD} \times 100 \tag{1}$$

where: PD—post-emergence plant density (no m$^{-2}$), SD—seeding density (grain m$^{-2}$).

(6) The fuel inputs for the performance of soil tillage practices, the application of fertilizers into soil and seeding of seeds were read from the agricultural tractor computer fuel consumption display during effective operation. (7) The plant yields in each year of research were expressed for the standard content of water in the grain of cereals/rapeseed seeds, 15%/8%, respectively.

*2.5. Data Analysis*

The results of the field experiments were statistically verified. The ANOVA variance analysis was made. The significance of the effect of the experiment treatments was evaluated with test F, and the differences between the mean values of respective characteristics—with the Tukey post-hoc test at $p = 0.05$. The variation in the plant emergence at the production plantation scale is provided in the box and whisker diagram as standard error and standard deviation from the mean value. The results analysis involved the use of Excel Microsoft 2016 and Statistica 12 [55].

## 3. Results

In each research year, immediately after winter, in the first half of April, the soil moisture accounted for 25–30% vol. and it did not depend on the tillage method (Figure 6A–C). The soil moisture with the advancement of the vegetation period was decreasing. However, since the second half of April to the end of July, the moisture of the soil tilled following the Mzuri Pro-Til technology was higher than following the CT one. Only in the higher rainfall periods the soil moisture was increasing, and it was similar irrespective of the tillage method, e.g., the second half of May 2014, the second half of June 2015 or the second half of July 2016.

In summer, the most water in soil occurred in the inter-rows after the harvest of winter wheat grown in the ST-OP (Table 2). The water content in the soil in the row zone was significantly lower and it was similar in the soil under CT. A spatial (row, inter-row) variation in the resistance to penetration and bulk density of soil was similar as the soil moisture. After six years of tillage following the ST-OP with Mzuri Pro-Til machines, the share of water-resistant soil aggregates was about 10 percentage points higher than after CT.

After six years, there was a significant variation in the content of organic carbon in soil. The total contents of carbon, extractable carbon, the fraction of carbon after decalcification, and carbon of humic and fluvic acids were significantly higher in the soil under ST-OP than under CT (Table 3).

ST-OP performed regularly for a few years increased the number of microorganisms and the biological activity of soil (Table 4). The total count of bacteria, cellulolytic microorganisms, Actinobacteria, filamentous fungi, earthworms, and the activity of dehydrogenase in the soil tilled with the use of Mzuri Pro-Til machines was significantly higher than in the soil under CT.

The emergence of winter wheat exposed to the ST-OP and CT evaluated over three years in Poland, Ukraine, and Lithuania was similar. However, the evenness of emergence for large plantations established for the Mzuri Pro-Til technology was greater. It is evident from a lower value of standard deviation of that feature (Figure 7).

A comparison of the fuel inputs for tillage, pre-seeding fertilization and the seeding of winter crops (wheat, barley, rapeseed) performed following three different technologies in Poland pointed to the advantage of ST-OP over RT, especially CT (Figure 8). It resulted in the saving of 22.7 L ha$^{-1}$ and 31.3 L ha$^{-1}$ of fuel, respectively.

The mean many-year yield of winter rapeseed grown in the ST-OP using Mzuri Pro-Til machines was significantly higher than in CT after performing plough tillage, especially after RT (Figure 9). Also, the winter wheat yield in the ST-OP was higher than after reduced tillage. The grain yield of winter barley, however, did not depend on the method of tillage and seeding.

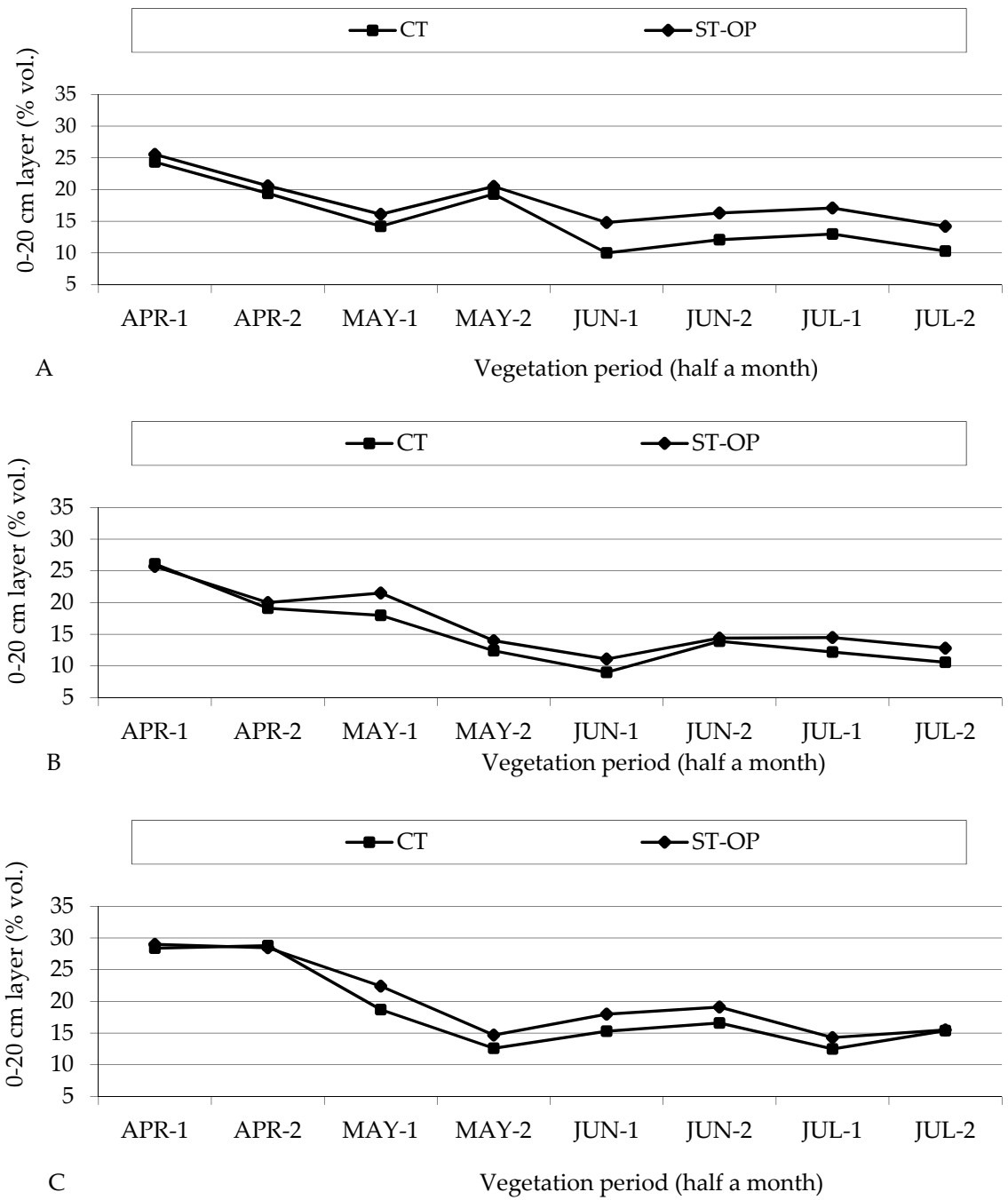

**Figure 6.** Soil moisture in the spring–summer period of plant vegetation depending on conventional tillage (CT) and strip-till one-pass technology (ST-OP) in (**A**) 2014, (**B**) 2015 and (**C**) 2016.

**Table 2.** Physical properties of soil after six years of varied tillage.

| Property | Strip-Till One-Pass | | Conventional | |
|---|---|---|---|---|
| | **Row** | **Inter-Row** | **Row** | **Inter-Row** |
| Moisture (% m/m) | 8.4 b | 10.5 a | 7.1 b | 8.0 b |
| Penetration resistance (MPa) | 2.26 b | 3.43 a | 2.14 b | 2.08 b |
| Bulk density (g cm$^{-3}$) | 1.41 b | 1.62 a | 1.32 b | 1.36 b |
| Water-resistant soil aggregates (%) | 40.8 a | 42.1 a | 32.5 b | 30.4 b |

a, b—letters in rows indicate significantly different at $P < 0.05$

**Table 3.** Content and fractions of organic carbon in soil after six years.

| Organic Carbon | Strip-Till One-Pass | Conventional |
|---|---|---|
| Total (g kg$^{-1}$) | 11.03 a | 9.85 b |
| Extractable (mg kg$^{-1}$) | 121.5 a | 103.1 b |
| After decalcification (mg kg$^{-1}$) | 366 a | 247 b |
| In humic acids (mg kg$^{-1}$) | 3073 a | 2588 b |
| In fluvic acids (mg kg$^{-1}$) | 3054 a | 2511 b |

a, b—letters in rows indicate significantly different at $P < 0.05$

**Table 4.** Biological properties of soil after the winter rapeseed harvest.

| Property | Strip-Till One-Pass | Conventional |
|---|---|---|
| Total count of bacteria ($10^6$ cfu g$^{-1}$ d.m. of soil) | 28.6 a | 20.4 b |
| Count of cellulolytic microorganisms ($10^6$ cfu g$^{-1}$ d.m. of soil) | 19.8 a | 15.8 b |
| Count of Actinobacteria ($10^5$ cfu g$^{-1}$ d.m. of soil) | 35.0 a | 31.7 b |
| Count of filamentous fungi ($10^4$ cfu g$^{-1}$ d.m. of soil) | 55.1 a | 34.3 b |
| Count of earthworms (no m$^{-2}$) | 36.5 a | 19.1 b |
| Activity of dehydrogenase (mg TPF kg$^{-1}$ d.m. of soil 24 h$^{-1}$) | 0.906 a | 0.711 b |

a, b—letters in rows indicate significantly different at $P < 0.05$.

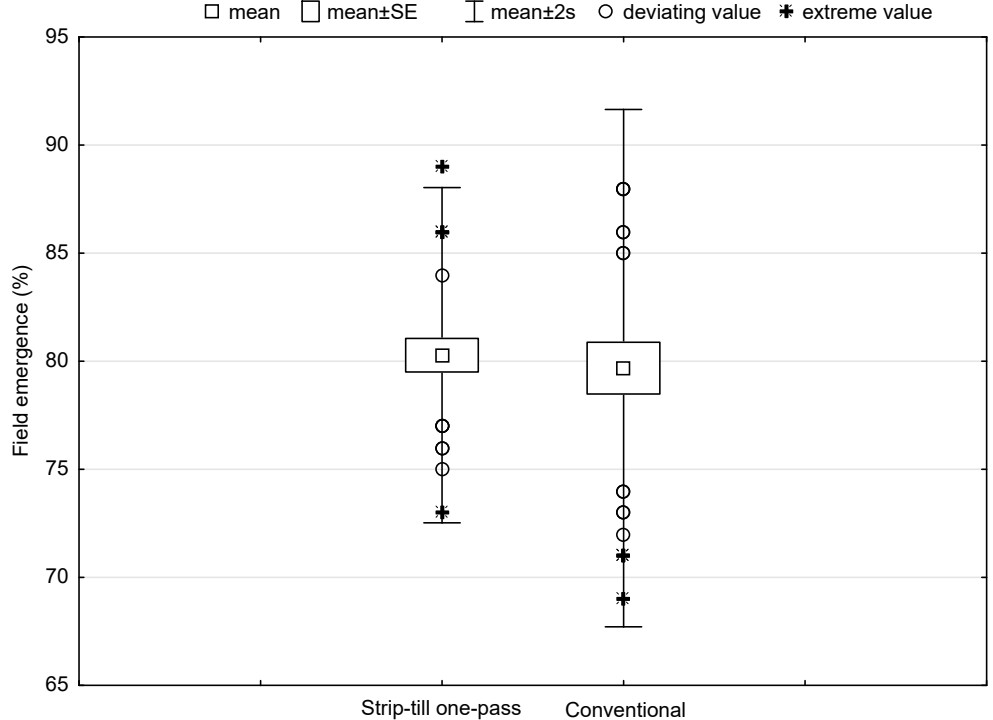

**Figure 7.** Winter wheat emergence on production plantations.

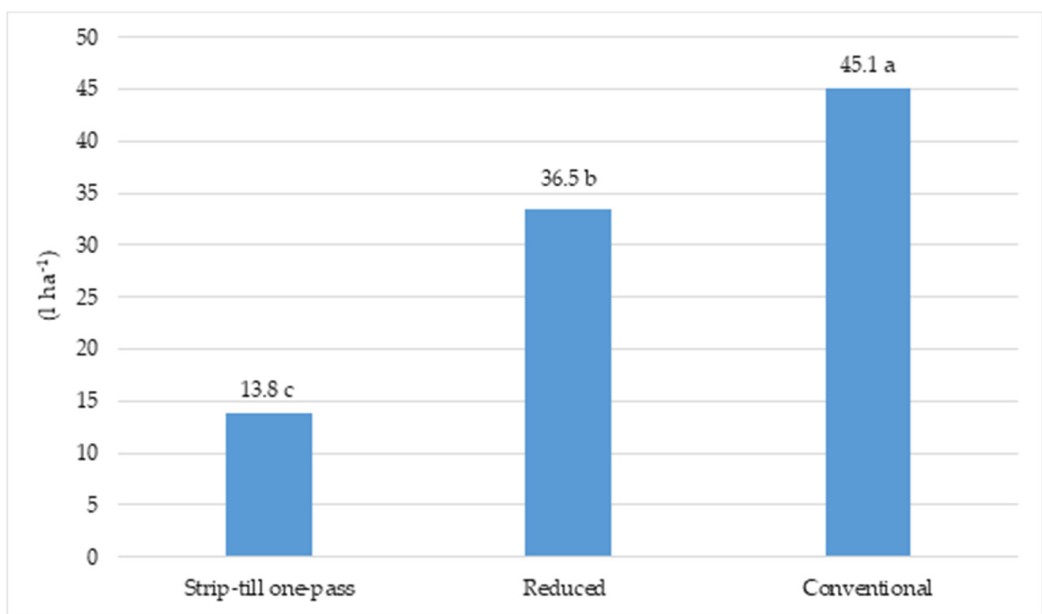

**Figure 8.** Fuel input depending on the technology of tillage, fertilization and seeding in winter crops (a, b, c —letters indicate significantly different at $P < 0.05$).

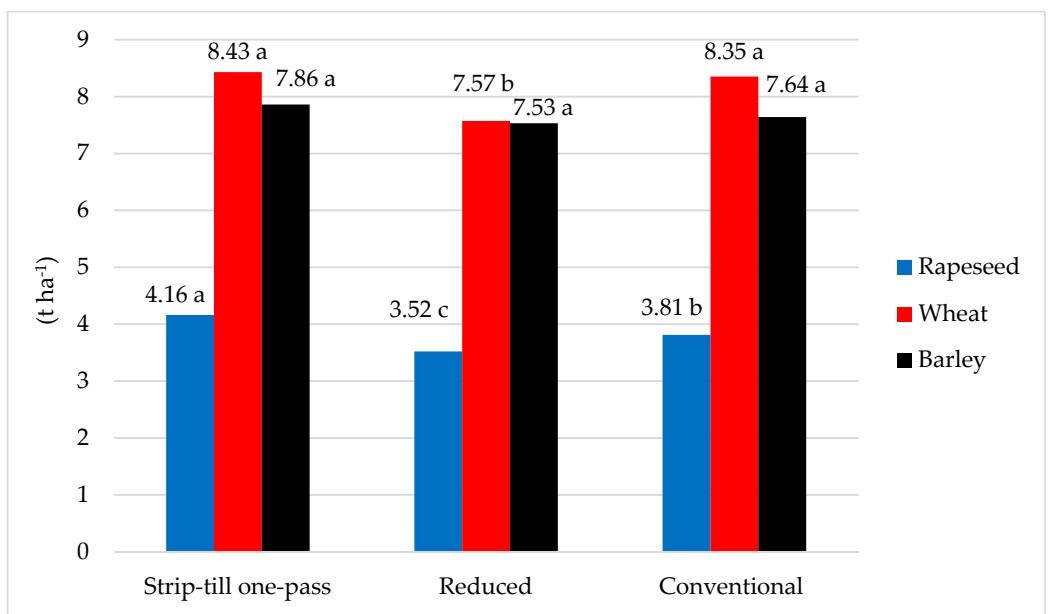

**Figure 9.** Winter crop yield depending on the technology of tillage, fertilization and seeding (a, b, c —letters indicate significantly different at $P < 0.05$).

## 4. Discussion

CT system dominates in Europe. Plough tillage method, however, requires high energy inputs and it is the source of greenhouse gas emissions [56]. For that reason, more and more frequently, similarly as in many parts of the world, different RT methods are applied. They are more environment-friendly and, after a few years of regular performance, the crop yields can be even higher than for CT [57]. ST loosens the soil less than CT and many reduced ploughless tillage methods. On the surface, there remains much crop residue, protecting the soil from erosion and runoff [58]. Such effect on soil allows for referring to the ST as conservation tillage [59]. In the countries where reduced tillage is common, ST is compared to NT [60]. In Poland and in other countries with a relatively low amount of precipitation,

the semi-drought periods reoccurring in the recent years, the occurrence of soil erosion, and with a negative soil organic matter balance, the alternative tillage methods and tools to CT system are being searched for.

The soil properties, spatial distribution of crop residue, and the conditions for plant growth are affected by the working parts of machinery and tools [61]. Tabatabaeekoloor [62] emphasizes that ST causes zonal differentiation of soil water and thermal conditions. Soil moisture in the loosened rows and non-loosened mulched inter-rows differs. Changes in the moisture of soil, both loosened and mulched, vary in time; they depend on the precipitation amount and distribution [63]. In the present research, the moisture of soil under CT and ST-OP was similar only in the periods of a high amount of water in the environment. It was after winter when evaporation was very low, where no transpiration occurred, and when much water from the melting snow reached the soil. According to Matula [64] and Lipiec et al. [65], in the non-loosened soil, e.g., in the inter-rows following the ST, water infiltration is limited. In loose soil, there is fast gravitational water mobility and evaporation. For that reason, in the vegetation period, after rainfall, the moisture of soil under CT and ST was getting even; however, already a few days later it was again higher under ST. Soil compaction, especially when accompanied by mulching, is important for water storage. It is stressed by, e.g., Wang et al. [66], investigating various tillage methods, including NT. In the present research, a zone-specific variation in the soil moisture after ST-OP persisted even after the crop harvest. A higher soil moisture was recorded in the inter-rows. That soil zone must have been less penetrated by the wheat roots than the row zone. Besides, on the surface, the previous crop residues were found, and mulching, as reported by other authors [67], is a good soil water protection method. The degree of soil loosening and moisture affect other physical properties of soil. For that reason, in the row zone the bulk density and penetration resistance were lower than in the non-loosened inter-rows. Those features in rows showed a similar value as the soil under CT. The zone tillage with a high amount of crop residue on the surface does not only protect the water in soil but also, when used for a few years, increases the content of organic matter in soil [68]. It also coincides with the results reported by Williams et al. [69]. In the present research, the content of organic carbon in the 0–20 cm after six years of tillage using Mzuri Pro-Til machines increased by 1.18 g kg$^{-1}$ of the soil, as compared with CT. Soil organic matter is an important component stabilizing the soil structure aggregates [70]. Al-Kaisi et al. [71], comparing the effect of five tillage systems, e.g., NT, ST, and CT on the soil structure, have found that the stability of micro- and macro-aggregates after many-year NT and ST was higher than after CT. The present many-year research also shows that after ST-OP the share of water-resistant aggregates increases, as compared with the soil under CT.

ST, excluding the soil inversion and limiting soil loosening to narrow strips, creates favorable conditions for soil microorganisms [72]. It results from a high content of organic matter in the surface layer, an optimal soil moisture and a stable aggregate structure. In the soils investigated in Śmielin, Poland, the ST-OP applied for a few years resulted in a significant increase in the count of various groups of soil microorganisms, earthworms, and in the enzymatic activity, as compared with CT.

In agricultural activities, more important than the absolute changes in respective soil properties is a reaction of crops to those changes. At the same time, it is expected that the reactions are the same on the surface of the entire field. It is possible only when there is no or if there is low variation in the soil conditions. In the fields of agricultural farms, it is rather impossible [73]. The variation in the physical, chemical, and/or biological properties of soil and the yield of crops is found not only in large-acreage fields, but also in small fields [74–76]. For that reason, each agrotechnical practice that decreases the variation in the conditions and plant production results at the scale of the field is desired. Earlier studies demonstrate that ST-OP can decrease the variation in the plant density and the yield of winter rapeseed and winter cereals in large arable fields, as compared with CT [52,53]. The observations and measurements of wheat density performed in three countries for three years confirmed that eliminating the inversion tillage, decreasing the number of soil-loosening practices, and applying the ST-OP decreased the spatial variation in winter wheat emergence on large plantations.

An even growth and development of crops in the field, ripening at similar time are important for the efficiency and effectiveness of the agrotechnical practices applied, the yield, and its quality [77].

Despite a documented favorable effect of ST-OP on the soil properties, the state of agricultural machinery market results in a scarce number of the scientific research results for cereals growing following this technology [78,79]. However, the technology triggers more and more interest in scientists and farmers. In many parts of the world, various methods of zone tillage and cereals seeding, especially wheat, are studied [80,81]. The grain yields are, in general, not lower, and even higher than in CT, RT or NT systems [82–84]. The mean three-year yield of winter rapeseed grown in Poland (Śmielin) was significantly higher than after CT and RT. Winter wheat yield was also higher than after the RT.

An essential argument for ST-OP are high fuel and labor savings, especially as compared with the CT most frequently applied in Europe. The observations and scientific research performed in the countries of Central and Eastern Europe confirmed those results. The analysis of the research results reported by Šarauskis et al. [85] shows that the highest inputs of labor time and fuel are required for the CT and sowing system. The application of reduced and NT systems can save even up to 2 h ha$^{-1}$. The fuel consumption in CT is more than five times higher than in NT system. According to Hosain et al. [86], the tillage of strips only can save 20% of fuel, as compared with RT of the entire field acreage. The use of Mzuri Pro-Til machinery for tillage, basic fertilization, and winter crop seeding in ST-OP resulted in saving even more than 30 L of diesel fuel per hectare, as compared with CT and more than 20 L ha$^{-1}$ with RT. A lower fuel consumption is also related to lower emissions of $CO_2$, which is stressed more and more [87]. Even though in the present research no direct $CO_2$ emissions have been determined, assuming that for the consumption of 1 litre of diesel fuel 2.75 kg $CO_2$ is produced [88], the tillage of 1 ha with Mzuri Pro-Til machines allows for reducing the emissions of that greenhouse gas by more than 50 kg as compared with RT, and by more than 80 kg as compared with CT.

## 5. Conclusions

Mzuri Pro-Til machines are present in four continents, especially in many countries of Central and Eastern Europe. The hybrid design facilitates the performance of agrotechnical practices in one-pass: tillage, fertilization, seeding, microgranulate application, catch crop seeding. It is possible to seed the crops in a narrow and wide-row spacing, the seeds are precision-distributed or placed in rows in one line, in two rows or in belts. The machinery model assortment allows for working with tractors of various class, in small- and big-acreage fields. The many-year application of ST-OP with Mzuri Pro-Til machines triggers favorable changes in the soil properties, as compared with intensive soil-inversion tillage. The soil contains more water, especially in the periods with low precipitation. The soil moisture in the precipitation deficit period is a few percentage points higher than after plough. After six years of a regular application of ST-OP, the content of organic carbon in the 0–20 cm soil layer increased by more than 1 g kg$^{-1}$, the content of organic carbon fraction (extractable, after decalcification, in humic and fluvic acids) was also higher. The soil recorded a higher count of bacteria, cellulolytic microorganisms, as well Actinobacteria, and more fungi and earthworms. The difference, as compared with the CT, was even more than 50%. In large fields with varied soil conditions, the plant emergence was even. ST-OP facilitated the yields not lower and frequently even higher than after CT or RT. A few-fold decrease in the consumption of fuel while maintaining the high yields has a favorable effect on the financial result of plant production. Besides, no soil inversion, loosening of the strips, and not the entire soil surface and mulching on the surface limit the greenhouse gas emissions. The advantages of the technology and machinery presented here justify a growing presence of that machinery in the countries with developing agriculture following the principles of sustainable development.

ST-OP should therefore be propagated under low-inputs agriculture conditions, especially in the regions of low precipitation, soils threatened with erosion and biological degradation.

**Author Contributions:** Conceptualization, I.J. and D.J.; methodology, I.J. and D.J.; investigation—performed the field experiments, D.J. and I.J.; data curation—compiled and analyzed the results, D.J. and I.J.; writing—original draft preparation, I.J. and D.J.; review and editing, D.J. and I.J. All authors have read and agreed to the published version of the manuscript.

**Funding:** This research received no external funding.

**Acknowledgments:** The authors thank the company Agro-Land Marek Różniak in Śmielin, Poland, for allowing experimentation in their production field with the use of the strip-till hybrid machine. Thanks also to Mzuri Pro-Til dealers in Central and Eastern Europe for the help in scientific observations and data collection.

**Conflicts of Interest:** The authors declare no conflict of interest.

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
