# Peer review of "Strip-Till One-Pass Technology in Central and Eastern Europe: A MZURI Pro-Til Hybrid Machine Case Study"

_agronomy, doi:10.3390/agronomy10070925_

Round 1

Reviewer 1 Report

The reviewed article presents valuable results of research and is worth publishing in ‘Agronomy’. Congratulations to Authors.

In general, the manuscript was carefully prepared. However, I have a few suggestions:

  • Lines 90-91: Please start this chapter with general information about the machine (e.g. like in line 82), not with the manufacturer details.
  • Section 2.2. The information presented in this section should be included to the Introduction. The description of the actual research area (countries and locations under experiments and observations, along with brief information on soil and climate conditions, if possible), which was the experimental basis for the article, should be included after the description of the strip-till one-pass technology.
  • Lines 154-156: The aim of the research should be presented at the end of the Introduction. Please compare lines 84-87 and make the aim of the research more precise.
  • Lines 157-170: Please specify which data come from static field experiments and which are the result of observations provided by Mzuri Pro-Til dealers. More information on field experiments in Åšmielin is needed; if the description of the experimental design is available in another published article, Authors may provide an appropriate link as supplementary material.
  • Line 169: “…the 4-year research period” – which years in the period of 2013-2019 were taken into account
  • Line 170: Please explain why these crops were examined (see the above suggestion).
  • Line 191: Please start as follows: “The evenness of winter wheat emergence was evaluated …” and then please specify the locations and years of investigations.
  • Line 195: Please supplement the years in which this research was conducted.
  • Tables 2-4 and Figures 8-9: Please add an explanation for the letters 'a' and 'b' shown next to the data, and specify which data should be compared in Table 2 (all figures in a particular row?)
  • Line 232: I suggest not using the term biodiversity, because the data in Table 4 does not justify it. 

I hope my suggestions will help Authors improve the article in terms of content and form.

Author Response

Thank you so much for Your comments.

Suggestions helped improve the article. The corrections were written in the manuscript (blue font). Some comments were similar to those of the second Reviewer. Corrections according to the comments of the second Reviewer were written in red.

The Introduction, Material and Methods chapters have been revised and supplemented. Due to the graphic representation of the research area and example of grown plants, chapter 2.2 has been separated. As suggested by the Reviewer, general environmental conditions for the experimental fields in Åšmielin (Poland) and literature sources in which they are described are given. The authors do not have accurate data from Central and Eastern European countries. Explanations for tables 2-4 and Figures 8, 9 have been made.

Reviewer 2 Report

Introduction.

Line 40. What authors have in mind when writing dominates? Perhaps the authors can justify what percentage of traditional plowing is now used in Europe? Strip tillage? No tillage?

The technological aspects of belt tillage are poorly described in the introduction. Which working parts have the greatest impact on the quality of belt tillage? What type of coulters and row cleaners are used in strip tillage machines. What row cleaners settings are best suited for this technology? What effect does the working depth of the coulters have on fuel consumption? Why is it not described what results other authors have obtained in previous studies?

Lines 66-67. How much fuel can be saved? How much is the potential for cost savings with strip tillage technology? How much can the yield increase? What plants? Why is it not described what results other authors have obtained in previous studies?

Table 1. Source?

Part of the methodology is described as in the manufacturers' brochures. Authors could describe in detail how the tests were performed, what the machine settings were, what the exact measurements were, and how they were performed?

Fig. 4. Why separation of plant residue is not combined with other machine parts?

What plants exactly were grown? Was there the same methodology in all countries? Was the technological process the same?

Lines 160-170. It is necessary to describe in detail the research methodologies applied to each point.

Lines 222-224. Why such results? Scientific substantiation required.

Line 240-243. What method was used to determine fuel savings. What results have been obtained by other researchers who have studied fuel consumption?

In Figures 7, 8 and 9, the technologies and plants are not the same.

The conclusions could be deeper and more scientific. What is the main conclusion of this work?

Author Response

Thank you so much for Your comments. As suggested, the information in the Introduction chapter has been clarified and supplemented (red font). The additions concern the results of scientific research, among others: - area of various soil cultivation systems, - working elements of belt removing machines, - fuel and cost savings. Some comments were similar to those of the second Reviewer. Corrections according to the comments of the second Reviewer were written in blue. At the request of the Reviewer, information on the research methodology was also supplemented. More details are provided about the conditions of field experiments, assessments and analyzes. The source of the information contained in Table 1 is given. The results shown in Fig. 7 come from different field experiments than the results shown in Fig. 8 and 9. The Conclusion chapter has also been supplemented and deepened.

Round 2

Reviewer 2 Report

The authors greatly improved their manuscript. Just before acceptance, I would recommend adding a more detailed description of the methodological methods (lines 205-221). It is necessary to provide for each point the exact test method, equipment and devices used (device number, manufacturer, country). For example, bulk density, soil aggregate water stability, total organic carbon and organic carbon fraction, etc. The clarity of the working methodology would be improved if the number of repetitions were given.

Author Response

Thank you so much for your comments and suggestions. The methodological methods (lines 205-221) have been described in more detail. These methods were also previously described on lines 222-250. This description has been completed. The corrections were written in green font.
